# API-Bank: A Comprehensive Benchmark for Tool-Augmented LLMs

**Minghao Li**[1*]**, Yingxiu Zhao**[2*]**, Bowen Yu**[1†]**, Feifan Song**[3]
**Hangyu Li**[1]**, Haiyang Yu**[1]**, Zhoujun Li**[4]**, Fei Huang**[1]**, Yongbin Li**[1]

[1]Alibaba Group, [2]Hong Kong University of Science and Technology,
[3]Peking University, [4]Shenzhen Intelligent Strong Technology Co., Ltd,
`{lmh397008, yubowen.ybw, hangyu.lhy}@alibaba-inc.com`
`{yifei.yhy, f.huang, shuide.lyb}@alibaba-inc.com`
`songff@stu.pku.edu.cn, yzhaocx@connect.ust.hk`
`lizhoujun@aistrong.com`

## Abstract

Recent research has demonstrated that Large Language Models (LLMs) can enhance their capabilities by utilizing external tools. However, three pivotal questions remain unanswered: (1) How effective are current LLMs in utilizing tools? (2) How can we enhance LLMs' ability to utilize tools? (3) What obstacles need to be overcome to leverage tools? To address these questions, we introduce API-Bank, a groundbreaking benchmark, specifically designed for tool-augmented LLMs. For the first question, we develop a runnable evaluation system consisting of 73 API tools. We annotate 314 tool-use dialogues with 753 API calls to assess the existing LLMs' capabilities in planning, retrieving, and calling APIs. For the second question, we construct a comprehensive training set containing 1,888 tool-use dialogues from 2,138 APIs spanning 1,000 distinct domains. Using this dataset, we train Lynx, a tool-augmented LLM initialized from Alpaca. Experimental results demonstrate that GPT-3.5 exhibits improved tool utilization compared to GPT-3, while GPT-4 excels in planning. However, there is still significant potential for further improvement. Moreover, Lynx surpasses Alpaca's tool utilization performance by more than 26 pts and approaches the effectiveness of GPT-3.5. Through error analysis, we highlight the key challenges for future research in this field to answer the third question [1].

## 1 Introduction

Over the past several years, significant progress has been made in the development of large language models (LLMs), including GPT-3 (Brown et al., 2020), Codex (Chen et al., 2021), ChatGPT, and impressive GPT-4 (Bubeck et al., 2023). These models exhibit increasingly human-like capabilities, such as powerful conversation, in-context learning, and code generation across a wide range of open-domain tasks (Bubeck et al., 2023).

Despite their utility, LLMs are still constrained by training data (Brown et al., 2020; Zhao et al., 2023; Song et al., 2023). This data can quickly become outdated and may not cover all scenarios (Mialon et al., 2023). Consequently, there has been a surge of research focused on enhancing LLMs by enabling them to leverage external API tools, such as accessing up-to-date information (Izacard et al., 2022) and interacting with third-party services (Liang et al., 2023). Traditionally, tool usage has been viewed as a uniquely human behavior, with its emergence considered a significant milestone in primate evolution (Ambrose, 2001). Drawing an analogy to the timeline of human evolution, we argue that it is currently imperative to address three pivotal questions: (1) How effective are current LLMs at utilizing tools? (2) How can we enhance LLMs' ability to utilize tools? (3) What obstacles still need to be overcome for LLMs to effectively leverage tools?

To tackle these inquiries, we present API-Bank, a groundbreaking benchmark specifically designed for tool-augmented LLMs. In order to determine the users' needs regarding the utilization of tool-augmented LLMs, we initially conducted interviews with 500 users. Taking into account their feedback, we establish the design principles for API-Bank. The evaluation scope of API-Bank must encompass three essential capabilities: *planning*, *retrieving*, and *calling* API tools, in order to fulfill the users' requirements. Additionally, while constructing the benchmark, it is imperative to consider various aspects such as domain diversity, API diversity, API authenticity, and evaluation authenticity. Subsequently, to answer the first research question, we implement the evaluation system of API-Bank, adhering to design principles. The sys-

---

[*] Equal Contributions
[†] Corresponding author.
[1]The data and code are publicly available at `https://github.com/AlibabaResearch/DAMO-ConvAI/tree/main/api-bank`

tem comprises 73 commonly used APIs, along with 314 tool-use dialogues containing 753 API calls, all manually annotated, thus forming the first executable and authentic system capable of evaluating the effectiveness of LLM utilizing tools.

For the second question, we develop a comprehensive tool-augmented LLM training dataset within API-Bank. This dataset comprises 2,138 distinct APIs and includes 1,888 dialogues with a total of 4,149 API calls, reflecting three different levels of API usage abilities. However, annotating thousands of APIs and the corresponding dialogues, adhering to our design principles, is not only prohibitively expensive but also time-consuming. In this work, we introduce Multi-agent, a novel method using LLMs instead of humans to automatically mass-produce tool-augmented LLM training data. Multi-agent consists of five collaborative agents that step-by-step generate domains, APIs, user queries, API calls & responses, while also ensuring the quality of the generated data aligns with our design principles. Multi-agent remarkably reduces the annotation cost by 98% compared to human annotation. To verify the effectiveness of our training set, subsequently, we fine-tune a popular LLM Alpaca-7B (Taori et al., 2023), resulting in our own tool-augmented LLM Lynx.

We conduct extensive experiments on API-Bank and obtained insightful results regarding the API tool usage capabilities of LLMs. Our findings reveal that even smaller models like Alpaca-7B and ChatGLM-6.2B possess a certain level of API call accuracy, about 20%. However, their proficiency in API retrieval and planning is negligible. In contrast, the larger-scale GPT-3 Davinci exhibits almost no capability in API usage, indicating that API usage might not be an inherent feature of LLMs. Our Lynx, demonstrates an average improvement of 24% in the three API usage capabilities compared to Alpaca-7B. While it approaches the performance of GPT-3.5, there remains a 21% gap from GPT-4. Additionally, we present a comprehensive experimental analysis that sheds light on the primary challenges encountered by GPT-4 and Lynx in utilizing APIs, addressing the third research question.

## 2 Design Principles of API-Bank

Due to the absence of authoritative ability definitions and benchmarks about tool-augmented LLM, we conducted an extensive questionnaire survey in the initial phase. Through interviews with over 500 users who expressed interest in incorporating additional tools into LLM, we collected their requirements. Based on this foundation, we provide, for the first time, a definition to measure the abilities of tool-augmented LLM and the data standards for training and evaluating its abilities. We believe that these design principles, which reflect the actual needs of users, can assist researchers in the field and contribute to the future development of tool-augmented LLM benchmarks.

### 2.1 Ability Grading

An ideal tool-augmented LLM should enable users to define the APIs they require in a private API Pool and request the LLM to invoke these APIs at the appropriate times to fulfill their needs. Based on our interviews with users, we have identified two dimensions that encompass the requirements for tool-augmented LLM:

- Few vs. Many APIs in Pool: Different users may define varying numbers of APIs. When users specify a small number of APIs (e.g., 2-3), the relevant information of all the specified APIs (such as name, definition, input parameters, and output parameters) and the user's query can be provided as input to the LLM, allowing it to determine which API to call to meet the user's needs. However, when users have defined a large number of APIs (e.g., hundreds), it is not feasible to input all the APIs to the LLM due to input length restrictions. In such cases, the LLM needs to retrieve potentially suitable APIs to fulfill the user's needs before calling them.

- Single vs. Several API calls per Turn: Many user requirements are complex and necessitate the combination of multiple API calls to achieve. Some users are willing to patiently decompose requirements into multiple queries and guide the LLM gradually to fulfill their needs, with each query involving a single API call. But some users prefer to directly present complex requirements, expecting the LLM to autonomously perform step-by-step API calls within a single turn to meet their needs.

As shown in Figure 2, these two dimensions give rise to four expected conditions: Few APIs, Single Call; Few APIs, Multiple Calls; Many APIs, Single Call; and Many APIs, Multiple Calls. During implementation, we discovered that the difficulty level

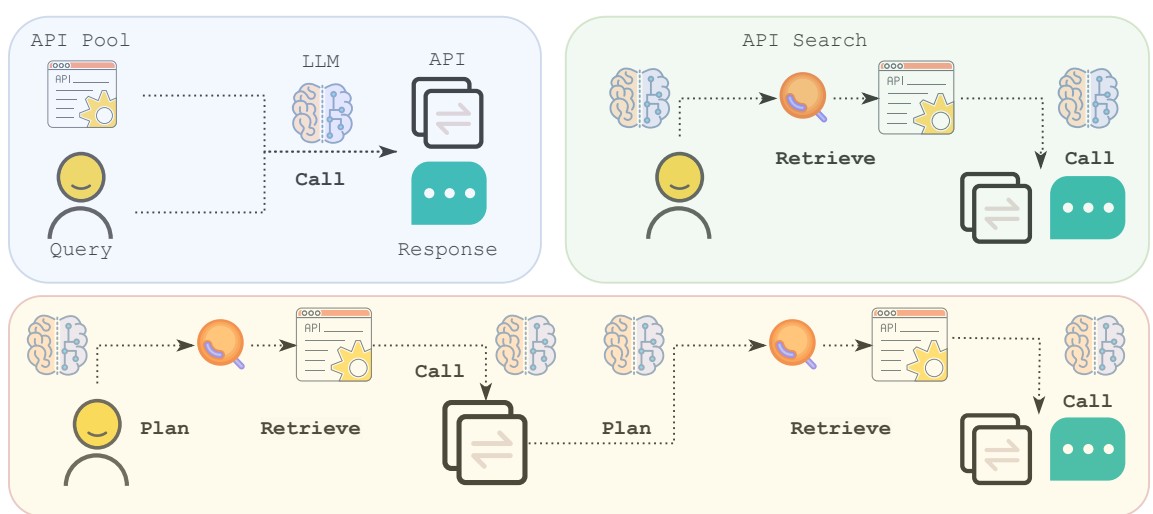

Figure 1: The abilities to *Call*, *Retrieve+Call*, and *Plan+Retrieve+Call* APIs of tool-augmented LLMs.

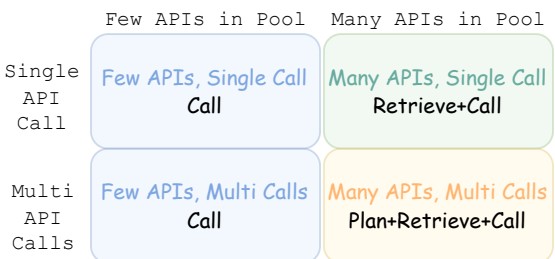

Figure 2: Requirements of tool-augmented LLMs.

of the first two conditions is similar. Since planning the call of multiple APIs is straightforward when all the APIs are input, thus we merged the first two conditions. The three remaining conditions assess the following abilities:

1. *Call*: the ability to call APIs based on the given query when the APIs are known;

2. *Retrieval+Call*: the ability to retrieve and call a single API when the APIs are unknown;

3. *Plan+Retrieval+Call*: the ability to continuously plan, retrieve, and call multiple APIs when the APIs are unknown.

We also present visual examples of each ability in Figure 1 and Appendix.

## 2.2 Data Standards

After grading the abilities to be assessed in the benchmark, another crucial aspect is ensuring the quality of the data. Calling APIs occurs within an open environment where we cannot predetermine the domains and functionalities. Moreover, API call is rigorous similar to mathematics, where

any errors during the call (such as incorrect API names, input parameter errors, or incorrect API call sequencing) can result in unfulfilled user requirements. Therefore, for the benchmark construction, we need to consider the following criteria:

1. **Domain diversity**: The training and testing data should cover a wide range of domains as comprehensively as possible;

2. **API authenticity**: The name, definition, input and output parameters of the APIs should closely resemble those in the real world;

3. **API diversity**: The benchmark should include a wide variety of API types and purposes.

4. **Evaluation authenticity**: The evaluation should incorporate a functional system that enables real-time interaction with the LLM. The LLM offers an API call, which the system executes and subsequently returns the results to the LLM. The evaluation is based on the impact of the execution on the system, assessing whether the LLM adequately responds to the user's requirements.

## 3 Evaluation System of API-Bank

Based on the aforementioned principles, we first introduce how to construct the evaluation system of API-Bank, including the system implementation, the data annotation, and the evaluation metrics.

### 3.1 System Implementation

We have implemented 73 APIs in our system, including commonly used daily APIs such as weather

forecast, and accessing other AI models like Text-to-Image Generation. All APIs were implemented within the same framework by senior research and development engineers, with a total time investment of 98 person-days. For APIs related to database operations, we establish the requisite databases and initialize them with initial entries, a pivotal step in constructing dialogues. For APIs that access external information (e.g., search engines), we must ensure the retrieved information remains constant to ensure reproducibility. We track all queries for each API in test dialogues and record the retrieval results at a specific time point, hard-coding them within the API to maintain result consistency.

Among them, we developed a special API called "API Search" to fulfill the evaluation requirements of both *Retrieval+Call* and *Plan+Retrieval+Call* abilities. Specifically, in these two scenarios, the LLM is unaware of the APIs available in the API Pool in advance, so it needs to make use of the API Search to identify the potentially needed APIs according to the user query. In the input given to the LLM, we provide the instructions of the API Search at the beginning, and an API Search is required before every other API call. When performing an API Search, the model should condense the user's demand into a few keywords. The API Search obtains sentence embeddings from both the query keywords and all API meta information in the API Pool. It calculates the cosine similarity between keywords and all API embeddings, returns the meta information of the API with the highest similarity.

### 3.2 Dialogue Annotation

Based on the grading abilities defined by design principles, we annotate evaluation data for the abilities of *Call*, *Retrieval+Call*, and *Plan+Retrieval+Call* APIs.

For the *Call* ability, we firstly randomly sample APIs from the API Pool. Then we instruct annotators to first imagine a query that could be resolved by these APIs based on the API document. They then annotate the API call and have the system execute it. Finally, they label the response based on the execution output. Please note that the *Call* data does not necessarily represent single-turn dialogues. We also require annotators to ask multiple queries regarding the same set of APIs, providing both the dialogue history and the API call history.

For the *Retrieval+Call* ability, we aim to obtain a complex user requirement and decompose it into multiple simple queries, each of which could be fulfilled by executing a single API. To achieve this, we initially obtain a set of APIs from the Pool, ranging from 1 to 5, and ask annotators to determine if they could collectively address a complex requirement. If so, they divide it into several simple queries. For each query, annotators label the API to be called by the LLM and provided the input parameters. They also label the response that the LLM should generate based on the system's output.

The annotation for *Plan+Retrieval+Call* is similar to that of *Retrieval+Call*, with the difference being that annotators don't decompose the complex query into simpler ones. They are required to annotate a sequential chain of API calls and the response derived from the execution of the last API.

The introduction of APIs increases the difficulty of annotation. Despite recruiting computer science students for dialogue annotation, each dialogue requires discussion between two annotators to decide how to annotate. Additionally, two additional annotators are involved to ensure the quality of annotations, including the format, logical consistency, and reasonability of API calls. The average annotation cost per dialogue is $8. Out of the 400 dialogues annotated, we discard 21.5% due to various annotation issues. Ultimately, we retained 314 dialogues with a total of 753 API Calls.

### 3.3 Evaluation Metrics

We evaluate model performance from two perspectives: the correctness of API calls and the quality of LLM responses. For the API call evaluation, we employ the Accuracy metric, which is calculated as the number of correct predictions divided by the total number of predictions. In each evaluation, we start by initializing the evaluation system, ensuring that the databases for each API contain default values. Then, we compare the predicted API calls with the manually annotated API calls to determine their consistency. We define consistency as whether the same database queries or modifications are performed and whether the returned results are the same. Regarding the evaluation of responses after API calls, we utilize the ROUGE-L metric.

## 4 Training Set of API-Bank

We hope API-Bank not only evaluate the effectiveness of existing LLMs in utilizing tools but also to enhance their performance in utilizing such tools.

The most direct approach to achieve this goal is creating a high-quality training dataset tailored to tool-augmented LLMs. However, it is challenging to construct a large-scale training dataset at a low cost while meeting the our design principles of domain diversity and API authenticity. The manual annotation cost for each dialogue in our evaluation set reaches $8, making it expensive to create a large dataset. Furthermore, it is challenging for the annotators to design a diverse and authentic API pool. Our recruited annotators could only come up with 100 APIs, which cannot satisfy our need for diverse training sets. Therefore, we propose a multi-agent data generation method to rapidly and cost-effectively construct tool-augmented LLM training data that aligns with the design principles.

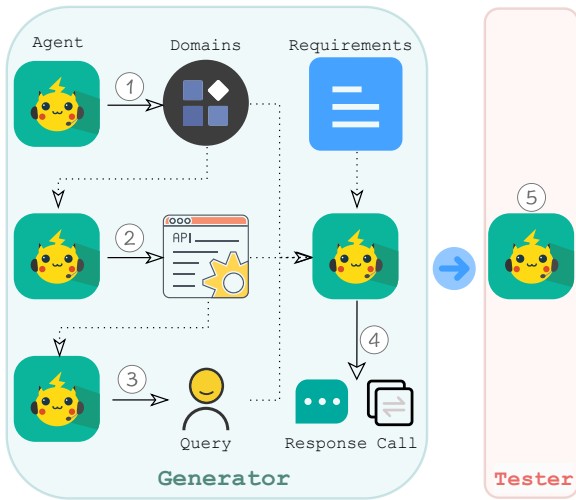

Figure 3: Five agents collaborate to accomplish the training data generation of tool-augmented LLMs.

The emergence of LLMs has brought about a paradigm shift in data annotation, offering an alternative to human annotators for the automated generation of labeled data. One representative approach is self-instruct (Wang et al., 2022). However, we have observed that while self-instruct is effective in generating data for simple tasks such as writing, it encounters difficulties in generating tool-augmented LLM data that aligns with our design principles. Specifically, we formulated a complex instruction encompassing requirements for domain diversity, API diversity&authenticity, and three specific abilities about *planning*, *retrieving* and *calling* APIs. Nevertheless, the widely used ChatGPT struggles to generate data that fully adheres to our instructions, only 5% of the data is available. Even upgraded to the more powerful GPT-4, the available rate improves to 25%, but a significant

number of errors still persist. Our investigation reveals that the errors stems from providing LLM with an overwhelming number of requirements all at once, making it difficult to comprehend them effectively. Consequently, an intuitive approach emerges: can we alleviate this issue by decomposing the requirements into multiple simpler tasks, allowing an LLM to execute one task at a time?

We commence by providing an overview of the elements that should be incorporated in the data based on our design principles: domain, API, query, ability, and API call&response. The domain determines the functionality of the APIs, while the APIs and ability determine the types of queries they can handle. The combination of domain, APIs, query, and ability determines how the LLM makes API calls and generates an appropriate response.

To simulate the dependencies among these elements, we propose the utilization of five agents as shown in Figure 3: (1) The first agent generates several domains, such as healthcare and fitness. (2) The second agent, considering the domain, generates potential APIs. It is worth noting that during this stage, to ensure the authenticity of the simulated APIs, we add examples from Public APIs[1] into the agent input. (3) The third agent randomly selects one or more APIs from the simulated APIs. Additionally, it chooses an ability outlined in our design principles. This information is then used to create a query that matches the chosen ability and can be fulfilled by calling the selected APIs. (4) The fourth agent takes the domain, API, ability, and query as input. It is expected to make the necessary API calls, simulate the execution of the APIs, and generate a response that addresses the query. (5) Finally, we introduce the fifth agent, acting as a tester. This agent automatically verifies whether the generated data adheres to our design principles (it actually discard 35% instances). All five agents are implemented by providing specific prompts to ChatGPT. Together, they collaborate in a step-by-step manner to accomplish the complex data generation. Multi-agent eliminates the need for human labelers, costing only $0.1 per dialogue and saving 98% compared to manual annotation.

## 5 Benchmark Analysis

**Statistics.** In the end, we construct a benchmark consisting of 1,008 domains, 2,211 APIs, 2,202 dialogues, and 6,135 turns. Among them,

---

[1]https://github.com/public-apis/public-apis

| Benchmark | Statistics | | Dialogue Type | | Evaluation Type | | Tool Usage Ability | | |
|---|---|---|---|---|---|---|---|---|---|
| | # domains | # APIs | Multi-turn | Multi-call | API Call | Response | Call | Retrieve | Plan |
| DATESET (Schick et al., 2023) | 1 | 1 | ✘ | ✘ | ✔ | ✘ | ✔ | ✘ | ✘ |
| APIBench (Patil et al., 2023) | 90 | 1,645 | ✘ | ✘ | ✔ | ✘ | ✔ | ✔ | ✘ |
| ToolAlpaca (Tang et al., 2023) | 50 | 426 | ✔ | ✘ | ✔ | ✔ | ✔ | ✘ | ✘ |
| ToolBench1 (Qin et al., 2023b) | 49 | 16,464 | ✔ | ✔ | ✔ | ✘ | ✔ | ✔ | ✔ |
| ToolBench2 (Xu et al., 2023) | 8 | 232 | ✘ | ✔ | ✔ | ✘ | ✔ | ✘ | ✘ |
| ToolQA (Zhuang et al., 2023) | 6 | 13 | ✘ | ✔ | ✔ | ✔ | ✔ | ✘ | ✘ |
| **API-Bank (ours)** | 1,000 | 2,138 | ✔ | ✔ | ✔ | ✔ | ✔ | ✔ | ✔ |

Table 1: Comparison among API-Bank and existing benchmarks, where API-Bank comprehensively includes the most domains and APIs, covers both multi-turn and multi-call dialogues, evaluate both API call and response, and thoroughly consider three distinct tool usage abilities.

| Statistics | Training | Evaluation |
|---|---|---|
| # of Domains | 1,000 | 8 |
| # of APIs | 2,138 | 73 |
| # of Dialogues | 1,888 | 314 |
| # of Turns | 5,221 | 914 |
| - # of single call | 3,147 | 363 |
| - # of multiple calls | 493 | 122 |
| # of Call | 720 | 214 |
| # of Retrieve+Call | 719 | 50 |
| # of Plan+Retrieve+Call | 449 | 50 |
| avg. turns per dialogue | 2.76 | 2.91 |

Table 2: Statistics of API-Bank.

there are 934 dialogues in the Call category, 769 in the Retrieve+Call category, and 499 in the Plan+Retrieve+Call category. Each dialogue has 2.76 turns in the training set and 2.91 turns in the testing set. The training data is generated automatically by the LLM, while the evaluation data is manually annotated. Therefore, variations exist among them in terms of domain, API scope, and dialogue content. We aim to assess the generalization of models trained on our benchmark by testing their performance on distribution-shift evaluation set. Please refer to Table 2 for detailed statistical data. We also provide samples and running demo of our benchmark in the Appendix.

**Quality.** Each instance in the evaluation set has been reviewed by four annotators, allowing us to primarily assess the quality of the training benchmark generated by Multi-agent. We randomly select 100 data samples from the training set and have the annotators evaluate their quality. The results demonstrate that the available rate of the Multi-agent generated data is 94%, which is an 89% improvement compared to data generated solely by a single agent (self-instruct), indicating that our training set possesses a high level of quality and reliability. Furthermore, upon examining the data that was automatically filtered by the tester agent,

we discover that 78% of it does not adhere to our design principles as intended. It shows that the tester agent can process data well according to the given principle.

**Comparison.** We have conducted a comparative analysis of API-Bank in relation to recently released benchmarks, as presented in Table 1. To ensure the comprehensiveness of this table, we have also incorporated some contemporaneous studies. Leveraging the design principles derived from our investigation of 500 user requirements, our API-Bank stands out among all benchmarks currently available for its: (1) Highest diversity: encompassing a wide range of domains and APIs. (2) Highest realism: simulating multi-turn dialogues that occur in real-world scenarios, including cases where multiple APIs are called in each turn. (3) Highest coverage: incorporating API call and response annotations, taking into account the essential capabilities required for tool-augmented LLMs, such as *planning*, *retrieving*, and *calling* APIs. Therefore, we firmly believe that API-Bank represents the most comprehensive benchmark for tool-augmented LLM available at present.

## 6 Related Work

Recent research in language modeling has explored the use of external tools to supplement the knowledge stored in the model's weights (Qin et al., 2023a). This approach allows for tasks such as exact computation or information retrieval to be offloaded to external modules such as a Python interpreter or a search engine (Mialon et al., 2023). These tools can include other neural networks or even the language model itself. Socratic Models (Zeng et al., 2022) is a modular framework that allows for the composition of different pre-trained models on various modalities. Alternatively, natural language knowledge can be retrieved from exter-

| Ability | LLM | Call | | Retrieve+Call | | Plan+Retrieve+Call | | Total | |
|---|---|---|---|---|---|---|---|---|---|
| | | Correctness | Rouge | Correctness | Rouge | Correctness | Rouge | Correctness | Rouge |
| Zero-shot | Alpaca-7B | 24.06% | 0.0204 | 5.19% | 0.0019 | 0.00% | 0.086 | 15.19% | 0.0318 |
| | ChatGLM-6B | 23.62% | 0.2451 | 13.33% | 0.2173 | 0.00% | 0.1522 | 16.42% | 0.2191 |
| | GPT-3 Davinci | 0.50% | 0.1035 | 1.48% | 0.091 | 0.00% | 0.0156 | 0.57% | 0.0814 |
| | GPT-3.5-turbo | 59.40% | 0.4598 | 38.52% | 0.3758 | 22.00% | 0.3809 | 47.16% | 0.4267 |
| | GPT-4 | 63.66% | 0.3691 | 37.04% | 0.351 | 70.00% | 0.4808 | 60.24% | 0.3910 |
| Fine-tuning | Lynx-7B | 49.87% | 0.4332 | 30.37% | 0.2503 | 20.00% | 0.3425 | 39.58% | 0.3794 |

Table 3: Main results of different LLMs on the API-Bank evaluation system.

nal sources, as demonstrated by WebGPT (Nakano et al., 2021) and ReAct (Yao et al., 2022) through the use of search APIs. Other approaches, such as Toolformer (Schick et al., 2023), ART (Paranjape et al., 2023), ToolkenGPT (Hao et al., 2023) and TRICE (Qiao et al., 2023) leverage a combination of search APIs, question-answering APIs, machine translation APIs, calculators, and other tools to solve various NLP tasks. ChatGPT Plugins [1] and TaskMatrix.AI (Liang et al., 2023) demonstrate the potential for language models to integrate with thousands to millions of APIs. LATM(Cai et al., 2023) and CREATOR (Qian et al., 2023) leverage GPT-4 to make API tools. Despite the promising demonstrations of these approaches, researchers have limited knowledge regarding three key issues: (1) How effectively can current LLMs utilize tools? (2) How can we enhance LLMs' ability to utilize tools? (3) What obstacles still need to be overcome for LLMs to effectively leverage tools? In this paper, we introduce API-Bank, the first benchmark specifically designed for tool-augmented LLMs, to address these three questions. As analyzed in Table 1, API-Bank is also the most diverse, realistic, and comprehensive tool-augmented LLM benchmark currently available.

# 7 Experiments

We proceed to fine-tune Lynx, a model based on LLaMA-7B (Touvron et al., 2023), using our API-Bank training dataset. The fine-tuning process consists of three epochs, with a batch size of 256 and a learning rate of 2e-5. Subsequently, we conduct a series of experiments on our API-Bank evaluation system, wherein we benchmark our model Lynx against other LLMs. The prompts used for evaluation are given in the Appendix. Through these experiments, we aimed to identify the remaining challenges that hinder the effective utilization of

LLMs in conjunction with tools.

## 7.1 Baselines

We evaluate the following models for our analysis: GPT-3 Davinci (Brown et al., 2020), the first powerful variant of the GPT-3 family of models. GPT-3.5-turbo with the gpt-3.5-turbo-0613 checkpoint; GPT-4, utilizing the gpt-4-0613 checkpoint; ChatGLM-6B (Du et al., 2022), a bilingual chatbot with 6B parameters; Alpaca-7B (Taori et al., 2023), an instruction-tuned variant of LLaMA-7B, leveraging 52K instruction-following data. Notably, Alpaca is currently recognized as the most widely used open-source LLM.

## 7.2 Main Results

We present the experimental results of currently public LLMs and our trained Lynx model on the API-Bank evaluation system, as shown in Table 3. As expected, the performance of each model gradually decreases with the increasing required ability. Given the API description, calling the API can be seen as a slot filling task. Both the 7B Alpaca and 6B ChatGLM achieve about 20% accuracy in API call, indicating that basic language models possess some tool-utilization ability. Surprisingly, GPT-3 Davinci, despite being a well-known 175B LLM, exhibits an extremely low correctness in this task. We speculate that this is because API call requires a strong understanding of instructions, which can only be unlocked through instruction tuning, a step omitted in GPT-3's training process.

On the other hand, the instruction-tuned GPT-3.5 demonstrates outstanding performance in this aspect, surpassing Alpaca-7B by 35 points in API call correctness and 0.44 in the response Rouge-L score. However, the effectiveness of GPT-3.5 decreases by 21% when compared to simple API calls in the *Retrieve+Call* setting, and a further 17% decrease is observed in the *Plan+Retrieve+Call* setting. This is because knowing which API to

[1]https://openai.com/blog/chatgpt-plugins

call and planning how to use the API both require certain reasoning abilities beyond instruction comprehension. GPT-4, currently known as the most powerful LLM, shows a 4 points improvement in API calls compared to GPT-3.5. Its performance in *Retrieve+Call* is similar to GPT-3.5, but it achieves a nearly 50% improvement in the most difficult *Plan+Retrieve+Call* setting. We hypothesize that this is due to GPT-4's emphasis on reasoning and planning abilities, similar to its performance in tasks involving mathematical reasoning.

An exciting discovery is that the Lynx model, trained on our API-Bank with Alpaca-7B initialization, outperforms Alpaca-7B by 26 points in API Call correctness and 0.41 in the response Rouge-L score, approaching the effectiveness of GPT-3.5. This strongly indicates the satisfactory quality of training data generated through our Multi-agent automated construction strategy.

Here, we would like to emphasize the differences between our developed API-Bank and the concurrently released APIBench(Patil et al., 2023) and ToolAlpaca(Tang et al., 2023) based on experimental results. GPT-3.5 achieves an API usage accuracy of 80% to 90% on their datasets. However, there is still significant room for improvement in our benchmark. This is because our evaluation set is manually constructed, considering design principles and closely resembling real-world scenarios. In contrast, their evaluation set is generated through self-instruct by the LLM and lacks diversity due to its narrow domain focus.

### 7.3 Error Analysis

| Error Type | Rate |
| --- | --- |
| No API Call | 36.77% |
| API Hallucination | 15.93% |
| Invalid Input Parameters | 7.96% |
| False API Call Format | 23.65% |
| Miss Input Parameters | 1.17% |

Table 4: Distribution of errors in the Alpaca evaluation.

In this study, we systematically categorize six primary error types, with detailed definitions provided in the Appendix. Notably, the primary issue identified during our evaluation of the original Alpaca model, as depicted in Table 4, is the frequent occurrence of "No API Call." However, it is worth mentioning that this issue exhibits a significant re-

| Error Type | Rate |
| --- | --- |
| API Hallucination | 61.38% |
| Has Exception | 16.40% |
| Invalid Input Parameters | 8.47% |
| False API Call Format | 6.88% |
| No API Call | 5.29% |
| Miss Input Parameters | 1.59% |

Table 5: Distribution of errors in the Lynx evaluation.

| Error Type | Rate |
| --- | --- |
| Failed API Retrieval | 67.86% |
| False API Call Format | 17.86% |
| Invalid Input Parameters | 7.14% |
| Miss Input Parameters | 7.14% |

Table 6: Distribution of errors in the GPT-4 evaluation.

duction in the Lynx model following fine-tuning with the API-Bank training dataset. This observed improvement suggests that disparities between the patterns of API calls in Alpaca's training dataset, which is constructed using 52,000 instruction data, and the actual API calls may contribute to this problem. It is plausible that the original Alpaca model encounters challenges in comprehending the API calls instructions in the evaluation system.

Another noteworthy issue we encounter is the presence of "False API Call Format." Given that our testing prompts are presented in a zero-shot format, the model has to rely solely on guidance about making API calls in instructions. This reliance on instruction-based guidance may cause difficulties for the model in learning the correct API call format. The application of fine-tuning results in a substantial enhancement in this regard, underscoring the significant benefit of fine-tuning in improving the model's capacity to generate accurately formatted API calls.

The analysis of the Lynx model results is given in Table 5, where the most significant type of error is the API name mismatch, accounting for 61% of the total errors. This occurs when the model makes API calls that do not match the annotated API calls in Ground Truth. In some cases, Lynx generates fake APIs that are unrelated to the user's intent, even if they are not provided in the test prompt. This issue arises due to Hallucination in Lynx, where it incorrectly calls APIs that it has encountered during training.

|  | Number of training samples | Accuracy (Call) | Rouge (Call) |
|---|---|---|---|
| ToolAlpaca | 10,366 | 53.88 | 39.75 |
| Lynx (Ours) | 6,184 | 54.64 | 39.80 |

Table 7: Results of the fine-tuned Alpaca on the ToolAlpaca dataset and Lynx.

The second major category of errors is related to problematic input parameters, which can lead to three types of errors: triggering an Exception within the API, being identified as an invalid input parameter, and generating an API call that fails to be parsed. These errors collectively contribute to 32% of the overall errors. Representative examples of problematic input parameters include passing a placeholder as a value, using an illegal format (such as not following the given date format, resulting in parsing failure), missing input parameters, and misunderstanding the requirements of the input parameter (e.g., requesting a stock code but passing in a company name). In addition, a few other issues exist, such as forging user or AI utterances instead of making an API call.

As shown in Table 6, the primary issue with GPT-4 is its inability to effectively use API search for retrieving specific APIs, which accounts for 68% of the overall problems encountered. This problem does not occur in the evaluation of the fine-tuned Lynx model. Fine-tuning appears to facilitate teaching the model the *Retrieve+Call* pipeline, unlike In-Context Learning, which makes it challenging to control the model's API calls as expected. The second common problem with GPT-4 is the API calls it gives cannot be parsed by the evaluation system. This issue arises because GPT-4 sometimes makes multiple simultaneous API calls, which violates the tested prompt's instructions.

Therefore, we aim to highlight three potential directions worth exploring: (1) Improved API calling methods: The direct generation of API calls, similar to Toolformer (Schick et al., 2023) , is limited by the number of available APIs. Introducing additional API retrieval tools is challenging due to issues such as hallucination and difficulties in accurately calling retrieval. (2) Enhanced API decoding algorithms: It is essential to demand strict adherence to the input parameter definitions when generating API calls with LLMs. (3) Larger-scale training data: While Lynx has demonstrated promising API usage capabilities, we wonder if scaling up the training data further could yield even better results.

We eagerly anticipate investigating this possibility.

## 7.4 Dataset Comparison

To assess the quality of the dataset created, we conducted experiments to fine-tune Alpaca using the training dataset of ToolAlpaca. ToolAlpaca automatically creates a tool-use corpus, which contains 3938 tool-use instances from more than 400 real-world tool APIs spanning 50 distinct categories. We convert the training set of ToolAlpaca into the training format consistent with ours, and finally obtained 10366 training samples. Since ToolAlpaca does not involve API *Retrieval* in its process, our evaluation focused only on its ability to *Call* APIs, ensuring fairness in the comparative analysis.

The results, presented in Table 7, reveal that our model Lynx, fine-tuned on the API-Bank dataset, outperforms the fine-tuned Alpaca on the ToolAlpaca dataset, even with fewer training data. This observation underscores the superior data quality of our constructed dataset and validates the efficacy of the multi-agent data synthesis methods.

## 8 Conclusion

This paper presents API-Bank, the pioneering benchmark for tool-augmented LLMs comprising 2,202 dialogues involving 2,211 APIs from 1,008 domains, and provides three key findings: (1) Current LLMs demonstrate some ability to use APIs, but there is still room for improvement in providing reliable services to users consistently. (2) Building diverse and high-quality training data is a promising approach to enhance LLMs' ability to use tools. (3) Proper API calls and strict adherence to API documentation pose significant challenges for LLMs in utilizing API tools. Overall, the API-Bank benchmark provides a groundbreaking resource for evaluating and advancing the state-of-the-art in tool-augmented LLMs. We believe that this research will inspire future studies and pave the way for developing more advanced AI systems that can intelligently integrate external resources to fulfill human requirements.

# 9 Limitations

API-Bank is a pioneering benchmark for tool-augmented LLM. While we have followed comprehensive design principles, it does have a few limitations. Firstly, API-Bank focuses solely on English, and we plan to address data construction and model evaluation for other languages as future work. Secondly, we have only fine-tuned Lynx-7B on API-Bank and have not explored larger-scale models, although Lynx-7B performs comparably to GPT-3.5. Lastly, we have trained a commercially viable tool-augmented LLM based on a larger LLM within our company. However, due to anonymity reasons, we are unable to report and analyze the results of the online model. We aim to include this information in future versions of our work.

# 10 Ethical Statement

This paper constructs a new benchmark for tool-augmented LLM, and we discuss some related ethical considerations here. Firstly, during interviews regarding their needs for tool-augmented LLMs, participants are informed in advance that their feedback will be used for product development and potentially published in the form of a research paper, without disclosing any personal private information. Secondly, all API tools in our dataset are original implementations and do not infringe upon any existing commercial software. Lastly, for data annotation in the benchmark, we hire four annotators. The hourly pay is set to 15 US$ per person, higher than the local statutory minimum wage.

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

# A  Appendix

> *Generate an API request in the format of [ApiName(key1='value1', key2='value2', ...)] based on the previous dialogue context.*
> *The current year is 2023.*
>
> **Input:**
> User: User's utterence
> AI: AI's response
>
> **Expected output:**
> API-Request: [ApiName(key1='value1', key2='value2', ...)]
>
> **API descriptions:**

Figure 4: The prompts used for API call evaluation.

> *Generate a response as an AI assistant based on the previous dialogue context and API request.*
> *The current year is 2023.*
>
> **Input:**
> User: User's utterence
> AI: AI's response
> API-Request: [ApiName(key1='value1', key2='value2', ...)]
>
> **Expected output:**
> AI: AI's response
>
> **API descriptions:**

Figure 5: The prompts used for response evaluation.

## A.1  Error Definitions

We briefly outline the definition of each type of errors:

- **API Hallucination**: The API name in the ground truth does not match the name in the prediction.

- **Has Exception**: The prediction triggers a custom or built-in Python exception, which should not be present in the ground truth.

- **Invalid Input Parameters**: The prediction contains invalid input parameters.

- **False API Call Format**: The prediction's API call format is not parseable.

- **No API Call**: There is no API call detected in the prediction.

- **Missing Input Parameters**: Necessary input parameters are missing from the prediction.

## A.2  Implement Details

The evaluation prompts consisted of two parts: one for API testing (Figure 4) and the other for response testing (Figure 5). We kept the prompts as concise as possible to showcase the models' fundamental capabilities.

## A.3  Examples

Because the huge amount of the domains in the training set, it is impossible to give a table statistics, so we only count the distribution of the API in the test set on the domain and give the following table:

| Domains | Numbers |
|---|---|
| Account Management | 7 |
| Information Query and Processing | 22 |
| Health Management | 8 |
| Schedule Management | 19 |
| Smart Home | 6 |
| Finance Management | 6 |
| Others | 5 |

Table 8: The domain statistics for the test set.

Here are some examples of the training and testing set domains along with their associated APIs:

**Training Set Domains:**
- Mental Health Hotline and Support
- Dental Procedure Cost Estimate
- Nutrition Planning

**Training Set APIs:**
- SearchDoctors
- GetPrice
- RecordMaintenance

**Testing Set Domains:**
- Account Management
- Health Management
- Entertainment
- Travel

**Testing Set APIs:**
- BookHotel
- DeleteAlarm
- SearchEngine

## Prompt

```
Generate an API request in the format of [ApiName(key1='value1', key2='value2', ...)] based on the previous dialogue context.
The current year is 2023.
Input:
User: User's utterence
AI: AI's response

Expected output:
API-Request: [ApiName(key1='value1', key2='value2', ...)]

API descriptions:
{"name": "DeleteAccount", "description": "Delete an account.", "input_parameters": {"token": {"type": "str", "description": "The
token of the user."}}, "output_parameters": {"status": {"type": "str", "description": "success or failed"}}}
{"name": "ForgotPassword", "description": "Sends an email to the user with a link to reset the password. Need call twice, first
with 'Forgot Password' status to get the verification code, then call again with 'Verification Code' status to change the
password. Must pass the name of the parameters when calling the API, like ForgotPassword(status='Forgot Password',
username='username').", "input_parameters": {"status": {"type": "str", "description": "'Forgot Password' for first call, after
get the verification code, call again with 'Verification Code' to change the password."}, "username": {"type": "str",
"description": "The username of the user. Only needed for the first call."}, "email": {"type": "str", "description": "The email
of the user. Only needed for the first call."}, "verification_code": {"type": "int", "description": "The verification code sent
to the user. Only needed for the second call."}, "new_password": {"type": "str", "description": "The new password of the user.
Only needed for the second call."}}, "output_parameters": {"status": {"type": "str", "description": "success or failed"}}}
{"name": "GetUserToken", "description": "Get the user token by username and password.", "input_parameters": {"username":
{"type": "str", "description": "The username of the user."}, "password": {"type": "str", "description": "The password of the
user."}}, "output_parameters": {"token": {"type": "str", "description": "The token of the user."}}}
{"name": "RegisterUser", "description": "The API for registering a account, given the username, password and email.",
"input_parameters": {"username": {"type": "str", "description": "The username of the user."}, "password": {"type": "str",
"description": "The password of the user."}, "email": {"type": "str", "description": "The email of the user."}},
"output_parameters": {"token": {"type": "str", "description": "The token of the user."}}}"
```

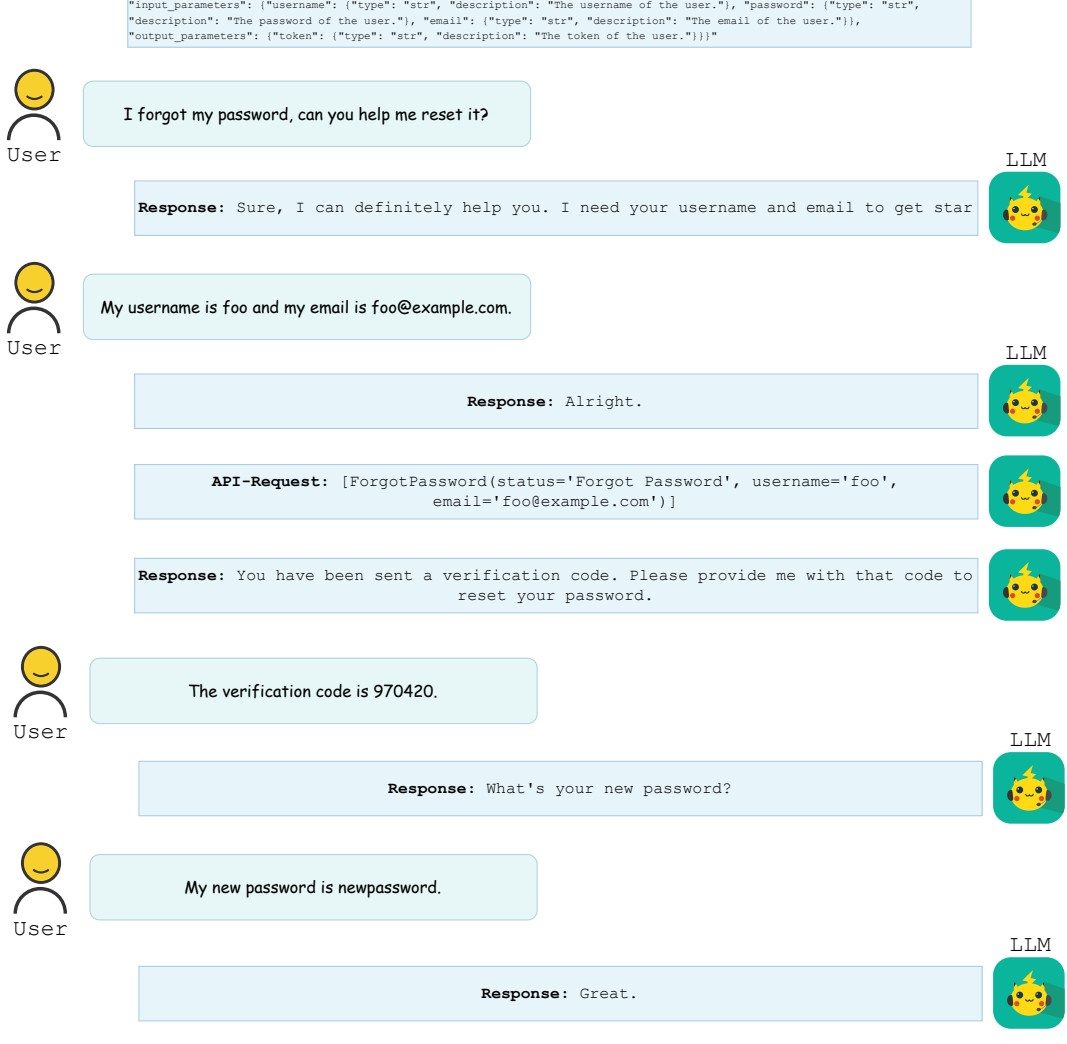

**User:** I forgot my password, can you help me reset it?

**Response:** Sure, I can definitely help you. I need your username and email to get star

**User:** My username is foo and my email is foo@example.com.

**Response:** Alright.

**API-Request:** [ForgotPassword(status='Forgot Password', username='foo', email='foo@example.com')]

**Response:** You have been sent a verification code. Please provide me with that code to reset your password.

**User:** The verification code is 970420.

**Response:** What's your new password?

**User:** My new password is newpassword.

**Response:** Great.

**API-Request:** [ForgotPassword(status='Verification Code', verification_code='970420', new_password='newpassword')]

**Observation:** Success

**Response:** Your password has been successfully reset.

Figure 6: One example of *Calling* APIs.

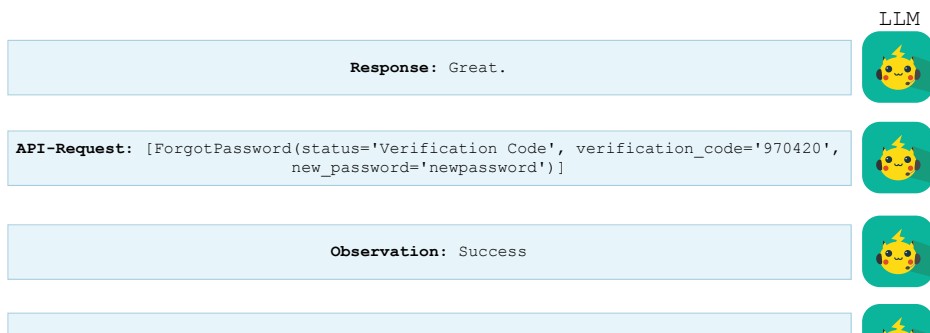

## Prompt

Generate an API request in the format of [ApiName(key1='value1', key2='value2', ...)] based on the previous dialogue context.
The current year is 2023.
Input:
User: User's utterence
AI: AI's response

Expected output:
API-Request: [ApiName(key1='value1', key2='value2', ...)]

API descriptions:
{"name": "ToolSearcher", "description": "Searches for relevant tools in library based on the keywords.", "input_parameters": {"keywords": {"type": "str", "description": "The keyword to search for."}}, "output_parameters": {"best_matchs": {"type": "Union[List[dict], dict]", "description": "The best match tool(s)."}}}

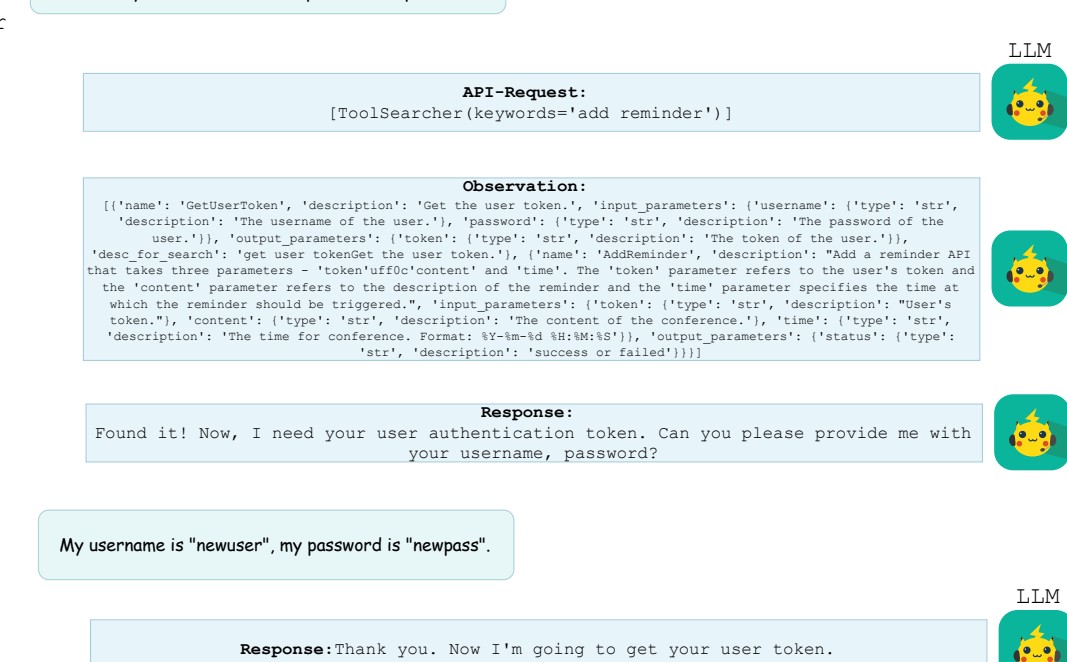

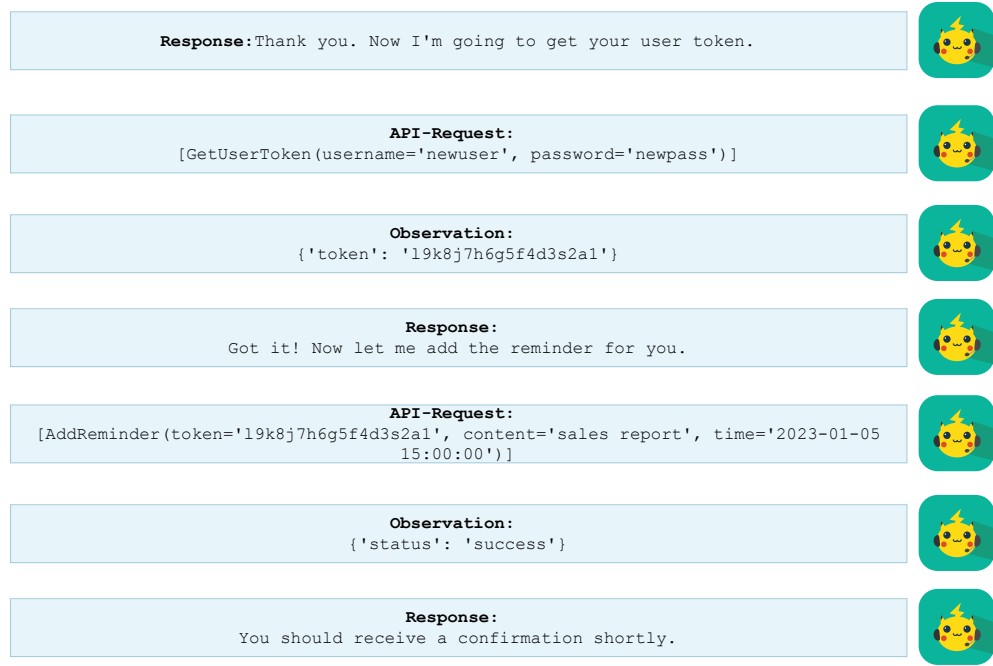

Figure 7: One example of *Retrieving+Calling* APIs.

## Prompt

You will be tested on your ability to make multiple API calls to fulfill a requirement based on a single sentence. You will be given an API box that includes a set of APIs such as a calculator, translator, WikiSearch, etc. When you want to use an API, you have to search for it in the API search engine using keywords. Try to describe it with these keywords. The tool search engine will then return you the most relevant information about the tool (api name, description, input/output parameters).

After you give each API call, stop generating and wait f3or input, I will return the results of the API call to you and make the next call based on the results. Your output is only an API call and does not contain any explanatory text, which means starts with [ and ends with ].
Here is an example of a test where ChatGPT represents you and API represents the return value.
Example:
Requirement: calculate the result of (5+3)*6 and sum the with 5.
ChatGPT: [ToolSearcher(keywords='calculator')]
API: {"name": "Calculator", "description": "This API provides basic arithmetic operations: addition, subtraction, multiplication, and division.", "input_parameters": {"formula": {"type": "str", "description": "The formula that needs to be calculated. Only integers are supported. Valid operators are +, -, *, /, and (, ). For example, '(1 + 2) * 3'."}}, "output_parameters": {"result": {"type": "float", "description": "The result of the formula."}}}
ChatGPT: [Calculator(formula='(5+3)*6')]
API: {'result': 48}
ChatGPT: [Calculator(formula='48+5')]
API: {'result': 53}

API description:
{"apiCode": "ToolSearcher", "description": "Searches for relevant tools in library based on the keywords.", "parameters": {"keywords": {"type": "str", "description": "The keyword to search for."}, "response": {"best_matchs": {"type": "Union[List[dict], dict]", "description": "The best match tool(s)."}}}

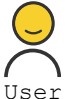
User

> Calculate the monthly salary after taxes for a
> Financial Analyst

LLM

**API-Request:**
[ToolSearcher(keywords='GetOccupationSalary')]

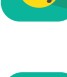

**Observation:**
{'api_name': 'ToolSearcher', 'input': {'keywords': 'GetOccupationSalary'}, 'output': {'name': 'GetOccupationSalary', 'description': 'API for querying the salary of a given occupation.', 'input_parameters': {'occupation': {'type': 'str', 'description': 'The occupation to query.'}}, 'output_parameters': {'salary': {'type': 'float', 'description': 'The salary of the given occupation.'}}}, 'exception': None}

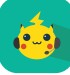

**API-Request:**
[GetOccupationSalary(occupation='Financial Analyst')]

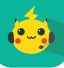

**Observation:**
{'api_name': 'GetOccupationSalary', 'input': {'occupation': 'Financial Analyst'}, 'output': {'salary': 100000}, 'exception': None}

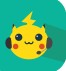

**API-Request:**
[ToolSearcher(keywords='TaxCalculator')]

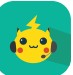

**Observation:**
{'api_name': 'ToolSearcher', 'input': {'keywords': 'TaxCalculator'}, 'output': {'name': 'TaxCalculator', 'description': 'API for calculating tax deductions based on the given salary.', 'input_parameters': {'salary': {'type': 'float', 'description': 'The salary to calculate tax deductions for.'}}, 'output_parameters': {'salary_after_tax': {'type': 'float', 'description': 'The salary after tax deductions.'}}}, 'exception': None}

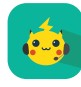

**API-Request:**
[TaxCalculator(salary='100000')]

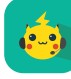

**Observation:**
{'salary_after_tax': 70000.0}

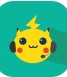

**Response:**
For a Financial Analyst with a salary of $100,000, the monthly salary after taxes is $70,000

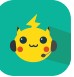

Figure 8: One example of *Planning+Retrieving+Calling* APIs.