# OpenReview forum: "API-Bank: A Comprehensive Benchmark for Tool-Augmented LLMs"
_EMNLP/2023/Conference — EMNLP 2023 Main_

### Official Review · Reviewer_vaMb · 2023-08-02

**Typos Grammar Style And Presentation Improvements:** N/A
**Soundness:** 4

**Excitement:**

4: Strong: This paper deepens the understanding of some phenomenon or lowers the barriers to an existing research direction.

**Missing References:**

See them in drawbacks and questions.

**Paper Topic And Main Contributions:**

The paper introduces a new benchmarking platform, API-Bank, to evaluate the efficacy of the tool-augmented large language models (LLMs). API-Bank is a comprehensive benchmark that includes 3-levels of data, answering three key questions in tool-use domain. These resources are designed to thoroughly evaluate the ability of LLMs to plan step-by-step API calls, retrieve relevant APIs, and correctly execute API calls to meet human needs.

The authors made the following contributions:
- The authors present the first benchmark tailored for tool-augmented LLMs. This benchmark includes a variety of API tools and annotated dialogues to evaluate the performance of LLMs.
- The authors use API-Bank to evaluate the performance of different closed-source and open-source LLMs.
- The authors provide a detailed error analysis and case studies to demonstrate the feasibility of tool-augmented LLMs for daily use, as well as to identify the primary challenges that future research needs to address.

**Questions For The Authors:**

- Although this paper is one of the very first works on creating an evaluation benchmark for evaluating tool-augmented LLMs, the authors at least need to mention or compare with the most recent tool-usage benchmarks for a complete EMNLP submission, including 2 ToolBench (https://github.com/OpenBMB/ToolBench, https://github.com/sambanova/toolbench) and 1 ToolQA (https://arxiv.org/abs/2306.13304). I know some of them might be concurrent work, but it would be better if you can make a comparison with them.
- I noticed a very interesting number that the authors keep emphasizing in the writing: "1008 domains". But I could not find any explanations/illustrations of what are the domains. And it is still quite weird for me that 2202 dialogs are distributed on 1008 different domains. I am wondering if there is a visualization on the domain distribution or are there 2 dialogs on the same domain?
- What are the differences between other tuned tool-augmented LLMs (ToolAlpaca, ToolLLaMA, Gorilla) and Lynx from a methodology angle?

From the reviewer: I know it is not possible to compare all the recent methods/concurrent works. But as this direction is developing fast, it would be better if you can compare your work with the other works I mentioned in the question section. I will also take this into consideration when judging whether to change the score or not. Thanks!

**Reasons To Accept:**

- The paper presents API-Bank, the first benchmark tailored specifically for tool-augmented Large Language Models (LLMs). To the best of my knowledge, API-Bank is the first comprehensive benchmark in this direction.
- The paper divided their data into different levels of difficulty and propose three different levels of data to answer three important questions in the tool-augmented LLMs research.
- The paper is well-written and easy to follow. Good visualizations are also included to help with people understanding.
- The paper conducts extensive experiments on both closed- and open-source LLMs, evaluating their efficacy of tool usage.

**Reasons To Reject:**

- Not enough baselines. The authors only compare with the zero-shot closed-source LLMs (gpt-3, gpt-3.5, gpt-4) and zero-shot open-source LLMs(alpaca, ChatGLM). A lot of prompting-based tool-augmented LLMs are not tested, like ReAct (https://arxiv.org/abs/2210.03629), Reflexion (https://arxiv.org/abs/2303.11366v3), Chameleon (https://arxiv.org/abs/2304.09842), etc. Comparing not enough baselines limits the paper to only evaluating the tool-usage abilities of LLMs with no additional design. The current experiments seem to be too weak for me.
- Although the paper is easy to follow, the writing still needs to be improved. From my perspective, the authors wanted to cover a lot of things and miss the emphasis on what they really want to claim.
- Lack of implementation details. The details of tool implementation are missing in this paper and so are some important components (e.g., retreiver, etc.).

**Reproducibility:**

4: Could mostly reproduce the results, but there may be some variation because of sample variance or minor variations in their interpretation of the protocol or method.

**Reviewer Confidence:**

4: Quite sure. I tried to check the important points carefully. It's unlikely, though conceivable, that I missed something that should affect my ratings.

---

> ### Author Rebuttal · Authors · 2023-08-28
>
> Thank you for your valuable feedback. Based on your comments, we will make several improvements to our paper:
>
> 1. (W1) Explore the impact of different prompt styles on performance.
> 2. (W2) Reorganize the content for improved clarity.
> 3. (W3) Provide more details on tool implementation.
> 4. (Q1) Extend Table 1 to include relevant works.
> 5. (Q2) Include statistical data distribution analysis.
> 6. (Q3) Clarify the differences between our approach and other adapted tool-enhanced LLMs.
>
> Meanwhile, we hope our answers can sufficiently address your concerns, and if that is the case, we kindly ask you to consider increasing the score.
>
> ### **W1: Not enough baselines ... The current experiments seem to be too weak for me.**
>
> We appreciate the reviewer's comment regarding the need for a broader set of baselines in our evaluation. We want to clarify our approach and rationale.
>
> In the context of the latest large language models, we recognize that prompt design is a crucial aspect of their development. While our primary goal was to assess the practical tool-usage abilities of these models, we also hope this benchmark that could serve as a guideline for evaluating different styles of prompts.
>
> To maintain a focus on fundamental baselines, we deliberately refrained from incorporating specific examples or complex prompt techniques during evaluation. This choice was made to showcase the models' core capabilities without relying on intricate prompt engineering.
>
> We acknowledge the existence of other tool-augmented LLMs like ReAct, Reflexion, Chameleon, and others, and we appreciate the reviewer's suggestion to include them in the evaluation. However, our intention was to provide a foundational assessment of tool-usage abilities, and we plan to explore comparisons with these models and their prompt styles in future work.
>
>
> ### **W2: Although the paper is easy to follow, the writing still needs to be improved ... on what they really want to claim.**
>
> We appreciate the reviewer's feedback on the paper's clarity and focus. We want to emphasize our core contributions:
>
> 1. We develop an authentic and interactive evaluation system with 73 API implementations and 314 annotated tool-used dialogues. We are the first to propose design principles and measure the ability of large language models to use tools from three different levels of difficulty: call, retrieval, and planning.
> 2. Utilizing the technique of multi-agent cooperation, we construct the first comprehensive training dataset for multi-turn dialogues with multi-call APIs, tailored for tool-augmented LLMs. This dataset consists of 2,138 APIs and 1,888 dialogue annotations.
> 3. We finetune the constructed dataset with Alpaca-7B to build our tool-augmented LLM called Lynx. The experiment results verifiy the superior performance of Lynx over other open-resource LLMs and even comparable to GPT-3.5. Our extensive analyzes of experimental results also shed light on key challenges in the field of tool-augmented LLMs.
>
> We apologize for any readability issues and appreciate the feedback. We will reorganize the content for improved clarity in the camera-ready version.
>
> ### **W3: Lack of implementation details ... (e.g., retreiver, etc.).**
>
>
> We appreciate the reviewer's attention to implementation details. We would like to provide more clarity on our tool implementation:
>
> Regarding the **73 APIs**: The entire interactive environment used for evaluation was implemented in Python and includes 73 runnable APIs. For APIs related to database operations (e.g., adding reminders - database insert, canceling meetings - database delete), we initialize the corresponding databases and generate initial entries for dialogue construction. For APIs that access external information (e.g., search engines), we must ensure the retrieved information remains constant to ensure reproducibility. We track all queries for each API in test dialogues and record the retrieval results at a specific time point, hard-coding them within the API to maintain result consistency.
>
> As for the **retriever implementation**, we adopted a fundamental approach. The model generates calls in the format "APISearch(keywords='[keyword1 keyword2 ...]')." The "keywords" section consists of multiple keywords separated by spaces. The retriever first extracts a sentence embedding from the entire string (including spaces) using the model: "sentence-transformers/paraphrase-MiniLM-L3-v2." Then, it extracts sentence embeddings for all API meta information (API Name + API description) in the API Pool. It calculates the cosine similarity between keywords and all API embeddings and returns the meta information of the API with the highest similarity.
>
> We hope this clarification addresses the concerns about implementation details. Thank you for highlighting this, and we will ensure that these details are presented clearly in the final paper.
>
> ### **Q1: Although this paper ... a comparison with them.**
>
> | Benchmark       | Statistics |        | Dialogue Type |            | Evaluation Type |          |
> |-----------------|------------|--------|---------------|------------|-----------------|----------|
> |                 | # domains  | # APIs | Multi-turn    | Multi-call | API Call        | Response |
> | API-Bank (ours) | 1,000      | 2,138  | Y             | Y          | Y               | Y        |
> | ToolBench1      | 49         | 16,464 | Y             | Y          | Y               | N        |
> | ToolBench2      | 8          | 232    | N             | Y          | Y               | N        |
> | ToolQA          | 6          | 13     | N             | Y          | Y               | Y        |
>
> In our camera-ready version, we will extend Table 1 as indicated and include references to the relevant works for a more thorough evaluation and comparison with the mentioned benchmarks. Thank you for highlighting these benchmarks, and we will ensure their inclusion to enhance the completeness of our submission.
>
> ### **Q2: I noticed a very interesting number ... on the same domain?**
>
> For the training set construction, the diversity of the domain determines the diversity of the generated tool-used APIs and dialogues. Given 1000 various occupations in the world, we instructed ChatGPT (gpt-3.5-turbo) to generate related domains and approporiate APIs that can be called to fulfill users' needs based on each occupation.
>
> The training set obeys uniform distribution, consisting of 1000 domains and 2138 APIs (2 or 3 APIs per domain). We achieve dialogue variety in our dataset by interchanging the order of these three APIs, which allows us to synthesize 2-3 distinct dialogues for each domain.
> The test set, also referred to our evaluation system, consists of 8 domains and 73 manually implemented APIs.
>
>
>
> Here is the statistics table for the test set.
>
> | Domains                          | Numbers |
> |----------------------------------|---------|
> | Account Management               | 7       |
> | Information Query and Processing | 22      |
> | Health Management                | 8       |
> | Schedule Management              | 19      |
> | Smart Home                       | 6       |
> | Finance Management               | 6       |
> | Others                           | 5       |
>
> We also list some examples for the constructed domains, APIs for training and test sets.
>
> | Training set domains (example)                                                          | APIs (example)                             |
> |-----------------------------------------------------------------------------------------|--------------------------------------------|
> | Mental Health Hotline and Support,  Dental Procedure Cost Estimate,  Nutrition Planning | SearchDoctors, GetPrice, RecordMaintenance |
>
> | Testing set domains (example)                                | APIs (example)                       |
> |--------------------------------------------------------------|--------------------------------------|
> | Account Management, Health Management, Entertainment, Travel | BookHotel, DeleteAlarm, SearchEngine |
>
>
> ### **Q3: What are the differences between other tuned tool-augmented LLMs ... from a methodology angle?**
>
> The differences between our approach and other adapted tool-enhanced LLMs (ToolAlpaca, ToolLLaMA, Gorilla) are as follows
>
> In terms of data generation:
> - **ToolAlpaca**: It involves two phases - collecting tool names from the internet and synthesizing tool documentation using LLMs. The training data is generated through multi-agent interactions between users, assistants, and tool executors.
> - **ToolLLaMA**: This approach utilizes 16,464 publicly available RESTful APIs. It employs a depth-first search decision tree method to traverse feasible combinations of APIs across various scenarios, generating training data.
> - **Gorilla**: Gorilla uses a self-instruct method, starting with 18 seed instruction-API pairs to synthesize its training data.
>
> In terms of training:
>
> - All approaches use LM training objectives combined with masking to train the models to generate the desired outputs.
> - The main difference lies in the fine-tuning prompts. API-Bank uses pure instruction zero-shot prompts. ToolAlpaca and ToolLLaMa use ReAct-style prompts and model output styles. Gorilla employs custom prompts that include API calls and explanations, along with the model's output style.
>
> We hope this clarifies the differences in methodology between our work and the mentioned LLMs. Thank you for your question, and we'll ensure that these distinctions are made clear in the final paper.

---

### Official Review · Reviewer_FRbH · 2023-08-05

**Soundness:** 4

**Excitement:**

4: Strong: This paper deepens the understanding of some phenomenon or lowers the barriers to an existing research direction.

**Paper Topic And Main Contributions:**

The paper constructs a system-level platform for evaluating LLM API calls, which includes over 2000 APIs, and establishes a set of environments for continuous evaluation. The only downside is that this platform is not accessible during the review period, making it difficult to assess its reproducibility and actual quality.

**Reasons To Accept:**

1. Tool Learning is an essential topic in the era of LLM.

2. The paper provides very valuable resources, especially the large number of APIs and the virtual evaluation environment.

**Reasons To Reject:**

1. Apart from APIs, a very important issue is how to ensure the quality of queries (i.e., queries that can naturally and reasonably call multiple APIs), but this paper actually discusses this relatively little.

2. Some details of model replication are not discussed in detail, such as how existing models are implemented to call APIs, which makes the evaluation results of the given models somewhat difficult to measure in terms of their scientific validity.

**Reproducibility:**

4: Could mostly reproduce the results, but there may be some variation because of sample variance or minor variations in their interpretation of the protocol or method.

**Reviewer Confidence:**

4: Quite sure. I tried to check the important points carefully. It's unlikely, though conceivable, that I missed something that should affect my ratings.

---

> ### Author Rebuttal · Authors · 2023-08-28
>
> Thank you for your valuable feedback. Based on your comments, we will make several improvements to our paper:
> 1. (W1) **Quality of queries**: We will add more details about how to ensure the quality of queries
> 2. (W2) **Model replication**: We will include a section that provides a comprehensive overview of the test conditions, including the prompts used, the parameters used for model generation, and any relevant information that affects the test environment.
>
> Meanwhile, we hope our answers have well addressed your concerns.
>
> ### **W1: Apart from APIs, a very important issue ... discusses this relatively little.**
>
> In our data synthesis process, we take several steps to guarantee both the diversity and quality of user queries.
>
> For the training set construction, the diversity of the domain determines the diversity of the generated tool-used APIs and dialogues. Given 1000 various occupations in the world, we instructed ChatGPT (gpt-3.5-turbo) to generate related domains and approporiate APIs that can be called to fulfill users' needs based on each occupation.
>
> Additionally, in our multi-agent data synthesis, we employ two dedicated agents to enhance data quality. One agent performs an analysis of synthesized dialogs, API parameters, and results, generating a comprehensive quality check report. A second agent then corrects the data based on this report. Through random sampling, we have verified the effectiveness of this process and observed significant improvements in data quality, eliminating the need for additional filtering.
>
> Furthermore, for the evaluation set, each instance has been reviewed by four annotators to ensure its quality. To further guarantee the quality of the training data, we randomly selected 100 samples from the training set and had the annotators check their quality.
>
> We believe these measures collectively contribute to the high quality and diversity of the queries in our dataset, and we appreciate the reviewer's attention to this critical aspect of our work.
>
> ### **W2: Some details of model replication are not discussed in detail ... in terms of their scientic validity.**
>
> The models under evaluation were categorized into two groups: those directly used for inference and those that underwent fine-tuning. Importantly, both groups used the same prompts during evaluation. This design choice was made to ensure fairness in the evaluation process.
>
> The prompts themselves consisted of two parts, one for API testing and the other for response testing. We kept the prompts as concise as possible to showcase the models' fundamental capabilities. We acknowledge that future work can explore prompt variations, such as adopting the ReAct Prompt mode, to potentially improve performance. We intend to add these implementation details in the camera-ready version of the paper.
>
> **Prompt for API Call Testing**
> ```
> Generate an API request in the format of [ApiName(key1='value1', key2='value2', ...)] based on the previous dialogue context.
> The current year is 2023.
> Input:
> User: User's utterence
> AI: AI's response
>
> Expected output:
> API-Request: [ApiName(key1='value1', key2='value2', ...)]
>
> API descriptions:
> ```
>
> **Prompt for Response Testing**
> ```
> Generate a response as an AI assistant based on the previous dialogue context and API request.
> The current year is 2023.
> Input:
> User: User's utterence
> AI: AI's response
> API-Request: [ApiName(key1='value1', key2='value2', ...)]
>
> Expected output:
> AI: AI's response
>
> API descriptions:
> ```

---

### Official Review · Reviewer_1pDE · 2023-08-07

**Typos Grammar Style And Presentation Improvements:** Typos
**Soundness:** 3

**Excitement:**

3: Ambivalent: It has merits (e.g., it reports state-of-the-art results, the idea is nice), but there are key weaknesses (e.g., it describes incremental work), and it can significantly benefit from another round of revision. However, I won't object to accepting it if my co-reviewers champion it.

**Paper Topic And Main Contributions:**

* This work's main contributions are focused on improving tool-augmented large language models (LLMs) through benchmarking. The 3 main research questions addressed by this paper are through the following three proposed approaches:

   (i) What is the state of current LLMs: Evaluation of current LLMs' capability of utilizing tools.;

   (ii) How to enhance their capability of utilizing tools: Development of a large dataset to benchmark the performance. Proposed a method to accelerate the annotation of the required dataset. Propose their model Lynx that is finetuning of the Alpaca model using their dataset.;

   (iii) Obstacles and gaps for further improvement in leveraging the use of tools by LLMs: Error analysis resulting in discussion of scope for future directions.

* For the evaluation of current LLMs, the authors develop a dataset consisting of 73 tools on 314 dialogues making a total of 753 API calls. This is made manually by humans.

* The evaluation is made on the following aspects: planning, retrieving and calling of the APIs. They incorporate this with the help of their design principles of the API-Bank. For this, they define two dimensions of evaluation: Few vs Many APIs in a pool & Single vs Several API calls. The presence of few APIs tests the LLM's ability to call; in the presence of many APIs the ability to retrieve+call can be assessed; finally, multiple calls in the presence of several APIs add to the assessment of planning as well.

* The training dataset created has 1,888 dialogues from 1000 domains and 2,138 tools making a total of 4,149 API calls.

* For annotation of the large training dataset, the authors propose and use a method using LLMs - Multi-agent. This method consists of 5 LLM agents collaborating to iteratively generate the domain, API call, select the user query, call API and respond.

* The paper uses the performances of Alpaca, ChatGLM, GPT3.5 and GPT4 to make comparisons with their fine-tuned model Lynx.

**Reasons To Accept:**

* The annotation scheme designed and implemented for the evaluation dataset is comprehensive and well thought out.

* The comparisons made by the authors show that their dataset has a significantly higher diversity of domains and the number of distinct APIs. Their dataset also includes multi-turn dialogue with multiple APIs being called multiple times. Additionally, the proposed design principles of segregating API calls into bins for planning, retrieving and calling allow better diversity of complexity in the dataset. All these features strongly add to the value of the dataset in my opinion. It would be valuable to have such a comprehensive dataset for the development of tool-augmented LLMs.

**Reasons To Reject:**

* The distribution of the API calls is missing. This would show if the dataset has any bias on the usage of the API calls. The training and test sets are not defined. The authors indicate the 73 tool APIs selected are commonly used but do not list the tools that this dataset is consisting of. This makes it difficult to evaluate if the benchmark is even useful.

* The writing of this paper is not up to the mark. The paper makes many strong statements without sufficient support. A few of them for instance are:

   - "a ground-breaking benchmark" is a strong statement on L009 and is repeated to be used in L636,

   - "comprises 73 commonly used APIs" on L078 has no supporting evidence nor any list of the APIs provided in the paper,

   - "Our investigation reveals that the errors stems from providing LLM with an overwhelming number of requirements all at once, making it difficult to comprehend them effectively" on L349 does not refer to any study given in the paper,

   - "GPT-4, currently known as the most powerful LLM" on L537 is another strong statement,

   - "We hypothesize that this is due to GPT-4’s emphasis on reasoning and planning abilities, similar to its performance in tasks involving mathematical reasoning." on L542 is an unwanted statement on GPT4's mathematical reasoning which again lacks support. Although GPT4 is irrelevant to the author's work such instances reflect the low quality of grounded writing presented in this research article.

   - "In contrast, their evaluation set is generated through self-instruct by the LLM and lacks diversity due to its narrow domain focus" on L564 refers to the comparison made with other benchmarks. This is insufficient as there is no explanation about the diversity and distribution of the 1000 domains that the dataset uses.

* The work showcases finetuning performances on only a single model, Alpaca. The paper misses to compare the effect of finetuning Alpaca on other similar datasets and comparing their performance. This would have allowed assessing the quality of the dataset created.

* Section 7.3: Error analysis begins with tables 4 and 5 based on some experimental setups that are not introduced in the writing earlier. This makes it difficult in understanding what exactly the errors correspond to. The errors mentioned in these tables are also not explained earlier. Later in the text, the writing explains some stats in those tables but lacks reference to the table. All these contribute to the bad writing quality dropping the standards of the work.

* The error analysis in section 7.3 is only on the fine-tuned Alpaca (Lynx model) and GPT4. However, since GPT4 is an unknown model, the analysis of its errors adds no significant value to the analysis. The motivation to analyze only the two among the complete set of models that were evaluated is not clear and not explained. It would have made sense to compare the error analysis of Alpaca with fine-tuning (Lynx) and without fine-tuning on their dataset. That would have provided insight on the quality of their benchmark. However, this is also missing.

* Although the approach for data creation is sophisticated and well-developed, the evaluation and analysis of the benchmark remains weak in terms of comparing with other benchmarks the effectiveness on the task of developing tool-augmented LLMs.

**Reproducibility:**

4: Could mostly reproduce the results, but there may be some variation because of sample variance or minor variations in their interpretation of the protocol or method.

**Reviewer Confidence:**

4: Quite sure. I tried to check the important points carefully. It's unlikely, though conceivable, that I missed something that should affect my ratings.

---

> ### Author Rebuttal · Authors · 2023-08-28
>
> We truly appreciate your constructive comments. We address your concerns as follows:
>
> **We have added more details to our paper:**
> 1. (W1) Include statistical data distribution analysis.
> 2. (W4) Provide specific explanations of error types.
> 3. (W2) Revise all the inappropriate statements.
>
> **In the experimental section, we have added:**
> 1. (W3/W6) Compare Lynx with Alpaca fine-tuned on other training data to validate the effectiveness of the training set.
> 2. (W5) Compare the error types between fine-tuned Alpaca (Lynx) and non-fine-tuned Alpaca.
>
> Meanwhile, we hope our answers can sufficiently address your concerns, and if that is the case, we kindly ask you to consider increasing the score.
>
> ### **W1: The distribution of the API calls is missing. ... This makes it difficult to evaluate if the benchmark is even useful.**
> For the training set construction, the diversity of the domain determines the diversity of the generated tool-used APIs and dialogues. Given 1000 various occupations in the world, we instructed ChatGPT (gpt-3.5-turbo) to generate related domains and approporiate APIs that can be called to fulfill users' needs based on each occupation.
> The training set obeys uniform distribution, consisting of 1000 domains and 2138 APIs (2 or 3 APIs per domain). The test set, also referred to our evaluation system, consists of 8 domains and 73 manually implemented APIs.
>
> Here is the statistics table for the test set.
>
> | Domains                          | Numbers |
> |----------------------------------|---------|
> | Account Management               | 7       |
> | Information Query and Processing | 22      |
> | Health Management                | 8       |
> | Schedule Management              | 19      |
> | Smart Home                       | 6       |
> | Finance Management               | 6       |
> | Others                           | 5       |
>
> We also list some examples for the constructed domains, APIs for training and test sets.
>
> | Training set domains (example)                                                          | APIs (example)                             |
> |-----------------------------------------------------------------------------------------|--------------------------------------------|
> | Mental Health Hotline and Support,  Dental Procedure Cost Estimate,  Nutrition Planning | SearchDoctors, GetPrice, RecordMaintenance |
>
> | Testing set domains (example)                                | APIs (example)                       |
> |--------------------------------------------------------------|--------------------------------------|
> | Account Management, Health Management, Entertainment, Travel | BookHotel, DeleteAlarm, SearchEngine |
>
> ### **W2: The writing of this paper is not up to the mark. The paper makes many strong statements without sufficient support.**
> Your feedback is very valuable for us, and we appreciate your effort in pointing out those strong statements in our paper.
> We apologize for those statements that may have caused concerns. We understand and acknowledge the points you have raised, and we are committed to addressing all of them in the next version of our paper.
>
> ### **W3: The work showcases ... of the dataset created.**
>
> Kindly note that we present the first comprehensive training dataset for multi-turn dialogues with multi-call APIs, while matching the three levels evaluation of LLM abilities (Call, Retrieve, Plan).
>
> Considering your suggestion, we **conduct experiments to fine-tune Alpaca on the dataset build from ToolAlpaca**, ensuring a consistent prompt format with our API-Bank. Since ToolAlpaca does not involve api-retriever in its work, we only evaluate its ability of calling APIs to ensure the fairness for comparison.
>
> The results are shown in the following table. We can notice that our model Lynx training on API-Bank dataset achieves better performance than ToolAlpaca even with fewer training data. This also indicates the better data quality of our constructed dataset.
>
> |             | Number of training samples | Accuracy (Call) | Rouge (Call) |
> |-------------|----------------------------|-----------------|--------------|
> | ToolAlpaca  | 10366                      | 53.88           | 39.75       |
> | Lynx (Ours) | 6184                       | 54.64           | 39.80       |
>
>
> ### **W4: Section 7.3: Error analysis ... the standards of the work.**
>
> Sorry for the inconvenience caused by our poor writing. We will definitely refine our paper in the next version. Below, we briefly outline the definition of each type of Error:
> 1. **API Hallucination**: The API name in the ground truth does not match the name in the prediction.
> 2. **Has Exception**: The prediction triggers a custom or built-in Python exception, which should not be present in the ground truth.
> 3. **Invalid Input Parameters**: The prediction contains invalid input parameters.
> 4. **False API Call Format**: The prediction's API call format is not parseable.
> 5. **No API Call**: There is no API call detected in the prediction.
> 6. **Missing Input Parameters**: Necessary input parameters are missing from the prediction.
>
> ### **W5: The error analysis in section 7.3 ... However, this is also missing.**
>
> We compared Alpaca's error analysis with (Lynx) and without fine-tuning on the dataset with the following results:
>
> | Error Type               | Rate   | Num |
> |--------------------------|--------|-----|
> | No API Call              | 36.77% | 157 |
> | API Hallucination        | 15.93% | 68  |
> | Invalid Input Parameters | 7.96%  | 34  |
> | False API Call Format    | 23.65% | 101 |
> | Miss Input Parameters    | 1.17%  | 5   |
> **Alpaca Error Distribution**
>
> | Error Type               | Rate   | Num |
> |--------------------------|--------|-----|
> | No API Call              | 5.29%  | 10  |
> | API Hallucination        | 61.38% | 116 |
> | Invalid Input Parameters | 8.47%  | 16  |
> | False API Call Format    | 6.88%  | 13  |
> | Miss Input Parameters    | 1.59%  | 3   |
> **Lynx Error Distribution**
>
> The primary issue observed with the original Alpaca model in our evaluation was the occurrence of "No API Call." This issue was significantly reduced in the Lynx model after fine-tuning on the API-Bank training dataset. We speculate that differences in the patterns of API calls between Alpaca's training dataset (constructed using 52K instruction data) and the actual API calls may have contributed to this problem. Understanding the instructions for API calls in the training data may have been challenging for the model.
>
> Another issue we observed was "False API Call Format." Since our testing prompts were in a zero-shot format, relying solely on instructions to guide the model to make API calls might have made it difficult for the model to learn the correct API call format. Fine-tuning led to a significant improvement in this aspect, highlighting the substantial benefit of fine-tuning in enhancing the model's ability to produce correctly formatted API calls.
>
> ### **W6: Although the approach ... developing tool-augmented LLMs.**
>
> At the time of our research, there were no existing works that included training datasets specifically designed for tool-augmented LLMs. We were among the first to explore the methodology of synthesizing training data for this purpose. We also compared the performance of Tool-Alpaca against Alpaca as a point of reference.
> We would like to emphasize our key advantages:
> 1. **Complexity of Training Data**: In our training dataset, we synthesized multi-turn dialogues, each may containing multiple API calls. This complexity in the data is a distinguishing feature. It allows our model to handle situations where a single user query may involve multiple API calls. This complexity is not present in many existing benchmarks.
> 2. **API Call & Response Training**: Our training process not only involves teaching the model to make API calls but also to generate responses based on the API's return values. This comprehensive training approach ensures that our model has a deeper understanding of the entire API call process, including handling return values.
>
> We believe that these unique aspects of our benchmark set it apart and make it particularly well-suited for developing tool-augmented LLMs. While we acknowledge the importance of comparing benchmarks, the absence of comparable benchmarks during the time of our work limited our ability to make such comparisons. We hope that future research in this area can build upon our findings and extend these comparisons.

---

### Official Review · Reviewer_ZmrS · 2023-08-10

**Typos Grammar Style And Presentation Improvements:** Should add highlight in Table 3
**Soundness:** 4

**Excitement:**

4: Strong: This paper deepens the understanding of some phenomenon or lowers the barriers to an existing research direction.

**Missing References:**

References are adequate. Maybe a good idea to explore the prompt strategy you adopted and cite this paper: A Prompt Pattern Catalog to Enhance Prompt Engineering with ChatGPT

**Paper Topic And Main Contributions:**

This paper presents API-Bank, a benchmark for enhancing large language models' (LLMs) proficiency in using external tools and APIs. Key contributions include:
- Defining three core capabilities for tool-augmented LLMs: Call, Retrieve+Call, and Plan+Retrieve+Call APIs, forming a framework to assess tool usage.
- Implementing a functional evaluation system with 73 APIs spanning diverse domains. It incorporates 314 manually annotated dialogues featuring 753 API calls to benchmark existing LLMs.
- Introducing a training dataset generator using a multi-agent approach, significantly reducing annotation costs (98%) while maintaining diversity and authenticity.
- Training LLM "Lynx" on the API-Bank dataset. Experimental results show progress over untrained models but also disparities compared to human performance.
- Offering a comprehensive analysis of errors made by current state-of-the-art LLMs, unveiling challenges in API retrieval, decoding, and reasoning.

**Questions For The Authors:**

A. The reliance on ChatGPT for multi-agent data generation raises questions about the approach's resilience given ChatGPT's limitations. Is additional filtering necessary to maintain data quality?

B. Regarding human annotations, what measures were taken for quality control and agreement assessment among annotators? How was the overall annotation quality upheld?

C. The paper mentions a larger commercial tool-augmented LLM, can you share more details?

**Reasons To Accept:**

This work tackles the crucial task of enhancing LLMs with external tools and APIs, vital for their scalability. It introduces a comprehensive benchmark with clear metrics, realistic dialog evaluations, and an innovative multi-agent data generation method that cuts annotation costs. This technique could be valuable for complex LLM training data beyond this context. The model analysis offers unique insights into the current tool utilization challenges and potential research paths. By providing open access to benchmark data, code, and models, API-Bank bridges a notable resource gap in this field.

**Reasons To Reject:**

The training data generation primarily depends on ChatGPT, which could have limitations. Exploring other LLMs like Claude2 could be valuable. The paper only focuses on one model variant, Lynx-7B; using larger models might yield further insights. However, I do not think any of these issues warrant rejecting the paper. The benchmark and analysis greatly contribute to advancing this NLP field. I recommend accepting the paper.

**Reproducibility:**

4: Could mostly reproduce the results, but there may be some variation because of sample variance or minor variations in their interpretation of the protocol or method.

**Reviewer Confidence:**

4: Quite sure. I tried to check the important points carefully. It's unlikely, though conceivable, that I missed something that should affect my ratings.

---

> ### Author Rebuttal · Authors · 2023-08-28
>
> Thank you for your valuable feedback. Based on your comments, we will make several improvements to our paper:
>
> 1. (W1) Include a comparison of the data generation effects among other LLMs.
> 2. (W2) Explore the fine-tuned performance of different model variants.
>
> Meanwhile, we hope our answers have well addressed your concerns.
>
> ### **W1: Should explore other LLMs for data generation.**
> We acknowledge the potential value of exploring other LLMs such as Claude2 for data generation. The core requirement in the process of training data as well as in the practice of multi-agent is the ability to complete the task according to the given prompt. Chatgpt and Claude2 are recognized as two powerful LLMs with the ability to follow instructions. We have verified in this work that the idea of using ChatGPT for training data synthesis and the overall synthesis results are as expected, so it can be considered feasible to use Claude2 as the core for training data synthesis. We will compare the effectiveness of different LLMs used for data synthesis in future work.
>
> ### **W2: Should explore different model variants (larger models).**
> In addition, it is also valuable to explore the performance of different model variants at different scales for tool use after fine-tuning, but due to the time and resource constraints during the rebuttal period, we were not able to complete the corresponding model training and experimental validation, and we will explore the fine-tuned performance of different model variants later (e.g. 13B and 65B).
>
> ### **Q1: The reliance on ChatGPT for multi-agent data generation ... Is additional filtering necessary to maintain data quality?**
>
> In the multi-agent data synthesis, there are two additional agents dedicated to ensuring data quality. One agent analyzes the correctness of synthesized dialogs, API parameters, and results, generating a check report. A second agent corrects the data based on this report. We've verified the efficacy of this process through random sampling and found significant improvements in data quality. After examining, the agent-corrected data, we found that 78\% of the data that was automatically filtered by the tester agent did not adhere to our design principles as intended.
> To further guarantee the quality of the training data, we randomly selected 100 samples from the training set and had the annotators check their quality. The results show that the available rate of the Multi-agent generated data is 94\%, which is an 89\% improvement compared to data generated solely by a single agent (self-instruct), indicating that our training set possesses a high level of quality and reliability.
>
> ### **Q2: Regarding human annotations, what measures were taken for quality control ... How was the overall annotation quality upheld?**
> In our study, we took several measures to maintain annotation quality and agreement. We recruited computer science students for annotation but ensured that each dialogue was reviewed by two annotators for consistent annotation. Moreover, two additional annotators were employed to verify the quality of annotations, including format, logical consistency, and API call reasonability. Each instance has been reviewed by four annotators to ensure its quality.
>
> ### **Q3: The paper mentions a larger commercial tool-augmented LLM, can you share more details?**
> We appreciate your interest in the larger commercial tool-augmented LLM mentioned in our paper. However, due to company policy, we are unable to disclose specific details. In general terms, this commercial LLM is built upon an internal base model within the company. It allows individual users or business clients to register APIs and utilize zero-shot capabilities for making API calls. We regret that we cannot provide more detailed information, but we hope this general description offers some insight into its functionality.
>
> ### **Missing References & Typos Grammar Style And Presentation Improvements:**
> We appreciate your suggestion to include a reference to the paper titled "A Prompt Pattern Catalog to Enhance Prompt Engineering with ChatGPT" in our manuscript. We have reviewed the paper, and it indeed provides valuable insights into prompt engineering techniques for enhancing interactions with ChatGPT. Multi-Agent synthesis of training data and bootstrapping of models for API calls is also essentially Prompt engineering, and has a relatively strong relevance to this paper. We'll add a citation to this paper in the camera-ready version and also correct the typos problem you mentioned.

---

### Meta-Review · Area_Chair_MTbm · 2023-09-19

**Recommendation:** 4

**Metareview:**

This paper presents API-Bank, a benchmark for enhancing large language models' (LLMs) proficiency in using external tools and APIs. As mentioned by the reviewers, the paper make several contributions. For the evaluation of current LLMs, a dataset consisting of 73 tools on 314 dialogues (a total of 753 API calls) has been developed. It also includes multi-turn dialogue with multiple APIs being called.
The paper introduces a training dataset generator using a multi-agent approach, significantly reducing annotation costs while maintaining diversity and authenticity. The paper also proposes their model Lynx, which fine-tunes Alpaca model using their dataset.

The reviewers pointed towards some issues with the writing, along with missing details. For instance, the details of tool implementation are missing in this paper and so are some important components The distribution of API calls is missing. The authors have provided these details in the rebuttal, and it should be added to the main paper.

---

### Decision · Program_Chairs · 2023-10-07

**Decision:**

Accept-Main

**Comment:**

This paper presents API-Bank, a benchmark for enhancing large language models' (LLMs) proficiency in using external tools and APIs. As mentioned by the reviewers, the paper make several contributions. For the evaluation of current LLMs, a dataset consisting of 73 tools on 314 dialogues (a total of 753 API calls) has been developed. It also includes multi-turn dialogue with multiple APIs being called.
The paper introduces a training dataset generator using a multi-agent approach, significantly reducing annotation costs while maintaining diversity and authenticity. The paper also proposes their model Lynx, which fine-tunes Alpaca model using their dataset.

The reviewers pointed towards some issues with the writing, along with missing details. For instance, the details of tool implementation are missing in this paper and so are some important components The distribution of API calls is missing. The authors have provided these details in the rebuttal, and it should be added to the main paper.